# ANGPTL3 Variants Associate with Lower Levels of Irisin and C-Peptide in a Cohort of Arab Individuals

**DOI:** 10.3390/genes12050755

**Published:** 2021-05-17

**Authors:** Muath Alanbaei, Mohamed Abu-Farha, Prashantha Hebbar, Motasem Melhem, Betty S. Chandy, Emil Anoop, Preethi Cherian, Irina Al-Khairi, Fadi Alkayal, Fahd Al-Mulla, Jehad Abubaker, Thangavel Alphonse Thanaraj

**Affiliations:** 1Department of Medicine, Faculty of Medicine, Kuwait University, Safat 13110, Kuwait; muath.alanbaei@hsc.edu.kw; 2Biochemistry and Molecular Biology Department, Dasman Diabetes Institute, Dasman 15462, Kuwait; mohamed.abufarha@dasmaninstitute.org (M.A.-F.); preethi.cherian@dasmaninstitute.org (P.C.); irina.alkhairi@dasmaninstitute.org (I.A.-K.); 3Special Services Department, Dasman Diabetes Institute, Dasman 15462, Kuwait; motasem.melhem@dasmaninstitute.org (M.M.); betty.chandy@dasmaninstitute.org (B.S.C.); emil.anoop@dasmaninstitute.org (E.A.); 4Genetics and Bioinformatics Department, Dasman Diabetes Institute, Dasman 15462, Kuwait; prashantha.hebbar@dasmaninstitute.org (P.H.); kyal1@yahoo.com (F.A.); 5Research Division, Dasman Diabetes Institute, Dasman 15462, Kuwait

**Keywords:** ANGPTL3-DOCK7, irisin, c-peptide, triglyceride, interleukin 13, lipid metabolism, insulin resistance, Arab population

## Abstract

ANGPTL3 is an important regulator of lipid metabolism. Its inhibition in people with hypercholesteremia reduces plasma lipid levels dramatically. Genome-wide association studies have associated ANGPTL3 variants with lipid traits. Irisin, an exercise-modulated protein, has been associated with lipid metabolism. Intracellular accumulation of lipids impairs insulin action and contributes to metabolic disorders. In this study, we evaluate the impact of ANGPTL3 variants on levels of irisin and markers associated with lipid metabolism and insulin resistance. ANGPTL3 rs1748197 and rs12130333 variants were genotyped in a cohort of 278 Arab individuals from Kuwait. Levels of irisin and other metabolic markers were measured by ELISA. Significance of association signals was assessed using Bonferroni-corrected *p*-values and empirical *p*-values. The study variants were significantly associated with low levels of c-peptide and irisin. Levels of c-peptide and irisin were mediated by interaction between carrier genotypes (GA + AA) at rs1748197 and measures of IL13 and TG, respectively. While levels of c-peptide and IL13 were directly correlated in individuals with the reference genotype, they were inversely correlated in individuals with the carrier genotype. Irisin correlated positively with TG and was strong in individuals with carrier genotypes. These observations illustrate ANGPTL3 as a potential link connecting lipid metabolism, insulin resistance and cardioprotection.

## 1. Introduction

Dyslipidemia is a major metabolic disease that is affecting more people due to the increased trend in obesity rates worldwide. As a response to positive energy balance, excess energy is stored as triglyceride (TG) in adipose tissue as well as ectopically. While such a mechanism in humans helps in coping with fluctuating energy supplies, it predisposes persistently over-nourished individuals to weight gain, leading to obesity. This unfavorable lipid accumulation happens in “ectopic sites” such as the liver and skeletal muscle, and even in pancreatic β-cells [1]. Such “ectopic” lipid accumulation and disorders of fatty acid–lipid metabolism lead to metabolic dysregulation including increased insulin resistance, inflammation and oxidative stress [2]. Active research is being pursued to delineate the mechanistic details of the connection between lipid metabolism and insulin resistance and their modulation by physical exercise. 

Angeopoietin-like3 (ANGPTL3) protein plays an important role in controlling lipoprotein metabolism and it has been proposed that it may be a novel and effective target for the treatment of dyslipidemia and CVD [3,4,5]. In coordination with other ANGPTL proteins, mainly ANGPTL4 and 8, they collectively regulate lipoprotein lipase activity (LPL). LPL is an important regulator of lipid partitioning to different organs and its tight regulation is critical. ANGPTL3 can directly inhibit LPL or in combination with ANGPTL8 under feeding conditions, directing triglyceride to adipocytes for storage and away from ectopic sites. 

The *ANGPTL3* gene lies on the forward strand of chromosome 1 p31 within an intronic region of *DOCK7* gene, located on the reverse strand. It has been previously reported that a lead SNP, namely rs2131925, identifies *ANGTPL3* as a TG-associated locus. The rs2131925 is an intronic SNP in the *DOCK7* gene corresponding to a location upstream of *ANGPTL3*. The NHGRI-EBI GWAS Catalog [6], which is a curated collection of all published genome-wide association studies, revealed two more SNPs, rs1748197 and rs12130333 from *ANGTPL3*, having strong effects on lipid metabolism. Both these two variants are in strong linkage-disequilibrium (LD) with the lead SNP, the first one at r^2^ = 0.995 and the second one at r^2^ = 0.5. The rs1748197 is an intronic SNP from *ANGPTL3* gene and the rs12130333 is a regulatory region variant downstream of *ANGPTL3* gene. A number of reports in the literature present rs12130333 as located within the ANGPTL3 locus (i.e., *ANGPTL3* SNP) or as near *ANGPTL3* gene (e.g., [4,7,8,9,10]). Several annotations for trait associations were observed for these two variants. For instance, the rs1748197 was shown to be associated with serum metabolite levels of high-density lipoprotein (HDL) triglyceride (HDL-TG), monounsaturated fatty acid (MUFA), triglyceride in medium HDL particles (M-HDL-TG), phosphatidylcholine (PC), polyunsaturated fatty acid (PUFA), total choline (TotCho), total fatty acid (TotFA) and extremely large very low density lipoprotein (XXL_VLDL_C) [11]. While many epidemiological studies have shown that high plasma HDL-C concentrations are associated with lower cardiovascular risk [12,13], recent studies have shown that high HDL-TG is a factor that plays a role in increased cardiovascular risk associated with high TG and that this might be due to the dysfunctional HDL caused by TG overload that in turn could alter vascular healing mechanisms [14,15]. It is also concluded that HDL-TG is positively related to glucose and insulin levels (not to HbA1c), underlining its direct association with glucose metabolic disturbances, opposite to HDL-C and total HDL particles (HDL-P) [15]. On the other hand, the trait annotations observed for the study variant rs12130333 were its association with levels of TG and total cholesterol. Interestingly, the two study variants as well as the leading SNP at the ANGPTL3 locus for TG were associated with a decrease in LDL-C levels [16,17]. 

Based on data emerging from various GWA studies on ANGPTL3 and its role in regulating plasma lipid levels, ANGPTL3 has been targeted as a treatment for people with elevated plasma lipids, especially for people diagnosed with homozygous familial hypercholesterolemia. Evinacumab, a ANGPTL3 inhibitor, is a human monoclonal antibody that has been recently approved by United States Food and Drug Administration (FDA). In a Phase 3 clinical trial, patients receiving evinacumab showed a 49% reduction in LDL-C levels from baseline relative to those receiving placebo at week 24 after treatment [18]. Indeed, this result is very promising since LDL reduction in patients with homozygous familial hypercholesterolemia is very difficult [19]. However, the trial assessed only a small number of patients for a limited period where long-term safety and cardiovascular outcome risks were not fully examined [18]. 

Given the above reported link between the *ANGPTL3* SNP variants and lipid metabolism, our aim in this study was to identify associations between these variants and markers associated with lipid metabolism and insulin resistance, such as irisin, a recent marker that has been associated with insulin resistance and cardiovascular risk. Irisin was also shown to be primarily an exercise-induced protein. Irisin is a novel myokine product produced from cleaving a transmembrane protein named fibronectin type III domain-containing protein 5 (FNDC5). This cleaved form of FNDC5 is released into blood circulation and has been shown to play an important role in modulating browning of white adipose tissue during exercise [11]. We have previously reported an increased level of this protein in the plasma from obese and type 2 diabetic (T2 D) people [20]. There has been considerable interest in the recent past on delineating the physiological role of irisin in processes such as glucose homeostasis and metabolic disorders [21,22,23]. Furthermore, irisin is now termed as “exercise-induced myokine” and the role of physical exercise in inducing the level of irisin and thereby promoting healthy systematic metabolism is now being actively examined [24,25,26,27,28]. Irisin was also shown to be involved in fatty acid oxidation. Based on its involvement in various metabolic pathways, including lipid metabolism, we were interested in investigating the interaction between ANGPTL3 variants and plasma levels of irisin. 

## 2. Materials and Methods 

### 2.1. Recruitment of Participants and Study Cohort

The Ethical Review Committee of Dasman Diabetes Institute reviewed and approved the study protocol as per the guidelines of the Declaration of Helsinki and of the US Federal Policy for the Protection of Human Subjects (Study number RA2010-003). Native adult Kuwaiti individuals of Arab ethnicity were recruited as study subjects. Pregnant women were excluded. The cohort comprised 278 subjects. For each participant, upon enrolment we recorded data on age, sex, health disorders (e.g., diabetes and hypertension) and baseline characteristics such as height, weight, waist circumference and blood pressure. Information on whether the participant takes lipid-lowering or diabetes and antihypertensive medication was recorded and was subsequently used to adjust the models for genotype-trait association tests. Every participant signed the informed consent form before participating in the study.

### 2.2. Blood Sample Collection and Processing

After confirming that the participant had fasted overnight, blood samples were collected in EDTA-treated tubes. Gentra Puregene^®^ kit (Qiagen, Valencia, CA, USA) was used to extract DNA, which was quantified using Quant-iT™ PicoGreen^®^ dsDNA Assay Kit (Life Technologies, Grand Island, NY, USA) and Epoch Microplate Spectrophotometer (BioTek Instruments, Winooski, VT, USA). Absorbance values at 260–280 nm were checked for adherence to an optical density range of 1.8–2.1.

### 2.3. Estimation of Plasma Levels of Various Biomarkers

Plasma was separated from blood samples by centrifugation, aliquoted and stored at −80 °C. 

Plasma level of irisin was detected using the irisin recombinant enzyme immunoassay kit (Phoenix Pharmaceuticals, Inc., Burlingame, CA, USA. Cat #EK-067-29). For the assay, kit instructions were followed: briefly, plasma samples were thawed on ice and centrifuged for 5 min at 10,000× *g* at 4 °C to remove any remaining cells or platelets. Then samples were diluted 40× with the 1× assay buffer (provided in the kit). The intra-assay coefficient for this ELISA assay was 1.0–7.0%, while the inter-assay coefficient was < 20%. 

Plasma level of ANGPTL3 was detected with the Quantikine Human Angiopoietin-like 3 (ANGPTL3) ELISA (R&D systems, Minneapolis, MN, USA. Cat# DANL30) according to the manufacturer protocol. 

C-peptide plasma level was detected with Mercodia C-peptide ELISA (Mercodia AB, Sylveniusgatan, Sweden, Cat# 10-1136-01) according to the manufacturer protocol. 

### 2.4. Targeted Genotyping of the ANGPTL3 Study Variants rs1748197 and rs12130333

We performed candidate SNP genotyping using TaqMan^®^ Genotyping Assay kit on ABI 7500 Real-Time PCR System from Applied Biosystems (Foster City, CA, USA). Each polymerase chain reaction sample was composed of 10 ng of DNA, 5× FIREPol^®^ Master Mix (Solis BioDyne, Estonia), and 1 µL of 20× TaqMan^®^ SNP Genotyping Assay. We set the thermal cycling conditions at 60 °C for 1 min and at 95 °C for 15 min followed by 40 cycles of 95 °C for 15 s and 60 °C for 1 min. We performed Sanger sequencing, using the BigDye™ Terminator v.3.1 Cycle Sequencing on an Applied Biosystems 3730 xl DNA Analyzer, for selected cases of homozygotes and heterozygotes to validate genotypes determined by the above techniques.

### 2.5. Quality Procedures for SNP and Trait Measurements 

We used PLINK (version 1.9) [29] to assess SNP quality and statistical associations with traits. We calculated minor allele frequency (MAF) and Hardy–Weinberg equilibrium for the two study variants. Any quantitative trait value < Q1 − 1.5 × IQR or any value > Q3 + 1.5 × IQR was considered as an outlier and was excluded from further statistical analysis.

### 2.6. Allele-Based Association Tests and Thresholds for Ascertaining Statistical Significance

Allele-based statistical association tests for the two study variants with 28 quantitative traits and biomarker levels were performed using linear regression adjusting for regular corrections toward age and sex. We also adjusted for diabetes medication and lipid-lowering medication. Of the 28 quantitative traits, the matSpD tool [30] (available at sites.google.com/site/qutsgel/software/matspd-local-version, accessed on 20 January 2021) identified only 17 as effective independent traits. Correction for multiple testing was assessed by adjusting the *p*-value threshold for the 17 independent traits (0.05/effective number of independent traits), which was (0.05/17 = 0.003). Empirical *p*-values (*P_emp_*-value) were generated using the max (T) permutation procedure available in PLINK, based on 10,000 permutations. A threshold of <0.05 was set for *P_emp_*-value.

### 2.7. Assessing the Interaction of Correlations between Study Variants and Traits 

Interaction of association signal (involving a SNP and associating trait) with another trait (example: Irisin~ rs1748197 + age + sex + TG + rs1748197*TG) was evaluated using multivariable linear regression tests using R statistical environment [31]. Summary statistics on the relationships among the traits/biomarkers included percentage variation in response variable (r2), estimates denoting standardized β-coefficients, standard error and significance of test (P) for the genotype distributions. The *p*-value threshold was set at 0.05.

## 3. Results

### 3.1. Characteristics of the Two ANGPTL3 Variants

In our study cohort, comprising 278 participants, the minor allele (A) of rs1748197_G_A occurred at a frequency of 0.355 and the minor allele (T) of rs12130333_C_T occurred at 0.146. Both the SNPs passed the tests for Hardy–Weinberg equilibrium (HWE). Summary statistics on the two study variants are presented in Appendix A.

### 3.2. Characteristics of the Study Cohort 

The mean age of the participants in the study cohort was 46.25 ± 12.38 years, with the ratio of males to females as 1:1.22 (Table 1). The cohort comprised mostly class I obese people with a mean body mass index (BMI) of 29.93 ± 5.17 kg/m^2^ and mean waist circumference (WC) of 99.36 ± 13.36 cm. Up to 48.6% of the participants were obese and 43.16% had type 2 diabetes. Mean values for HbA1 c (6.31 ± 1.3%), LDL (3.13 ± 0.96 mmol/L), HDL (1.2 ± 0.32 mmol/L), total cholesterol (TC) (5.02 ± 1.09 mmol/L) and TG (1.22 ± 0.6 mmol/L) were normal or near normal. Of the 278 people, 101 were taking T2 D medication and 88 were taking medication to control the level of lipids in blood circulation.

Participants in our study cohort with diabetes differed significantly from those without diabetes in anthropometric trait measurements (age (*p*-value = 3.03 × 10^−^^12^), weight (*p*-value = 7.07 × 10^−^^6^), BMI (*p*-value = 1.36 × 10^−^^5^) and WC (*p*-value = 3.58 × 10^−^^9^)), lipid trait measurements (HDL (*p*-value = 0.0018) and TG (*p*-value = 7.09 × 10^−^^7^)), glycemic indices (FPG (*p*-value = 5.19 × 10^−^^16^) and HbA1 c (*p*-value = 2.2 × 10^−^^16^)), in obesity status, insulin (*p*-value = 0.051) and irisin (*p*-value = 1.83 × 10^−^^5^).

Examination of the clinical characteristics with participants distributed gender-wise (Appendix A) showed significant differences in the traits of height, weight, WC, HDL, TG and FPG.

Upon partitioning the cohort based on genotypes (GG versus GA + AA) at the rs1748197 variant, it was found that the sub-cohorts exhibited significant differences (*p*-value ≤ 0.05) in the mean values for irisin and c-peptide. In a similar manner, upon partitioning the cohort based on genotypes (CC versus CT + TT) at the rs12130333 variant, the sub-cohorts exhibited significant differences (*p*-value ≤ 0.05) in the mean values for c-peptide and irisin (Figure 1 and Appendix A). Significant differences were also seen in the levels of ANGPTL3 but only with the rs1748197 variant (40.077 ± 10.23 versus 35.65 ± 9.85; *p*-value = 0.004).

### 3.3. Association of the Two ANGPTL3 Study Variants with Lower Levels of c-Peptide and Irisin at Significant p-Values 

Statistically significant associations (*p*-value threshold corrected for multiple testing ≤ 0.0019 and empirical *p*-value, *P**_emp_*-value ≤ 0.05) between the variants (with the minor allele as the effect allele) and two traits (c-peptide and irisin) were observed (Table 2): rs1748197:c-peptide (*p*-value = 0.0001; *P**_emp_*-value =0.006; effect size = −0.6976); rs1748197: irisin (*p*-value = 0.0002; *P**_emp_*-value =0.01; effect size = −63.1); rs12130333:c-peptide (*p*-value = 0.0003; *P**_emp_*-value =0.015; effect size = −0.9002); rs12130333: irisin (*p*-value = 0.002; *P**_emp_*-value =0.098; effect size = −72.61). Though the *P**_emp_*-value is >0.05 for the last association (rs12130333: irisin), the *p*-value is significant. Both the variants were associated with lower levels of TG at *p*-value threshold not adjusted for multiple testing (≤0.05) (Appendix A). 

Evaluation of the identified association signals from these two variants with the levels of irisin and c-peptide revealed that these two association signals were consistently significant (*p*-value ≤ 0.05) in the sub-cohorts of individuals afflicted with diabetes (Appendix A).

### 3.4. Associations of the Haplotype of the ANGPTL3 Study Variants with the Levels of c-Peptide, and Irisin 

Upon considering the two study variants as a haplotype (rs1748197-rs12130333), we observed significant associations between the haplotype with irisin and c-peptide when the haplotype contains effect alleles at both the variants (rs1748197_A-rs12130333_T) or reference alleles at both the variants rs1748197_G-rs12130333_C) (Table 3). The association with irisin was particularly significant as both the *p*-values and *P**_emp_*-values were significant. While the reference haplotype (G-C) was associated with increased levels of irisin and c-peptide, the haplotype with effect alleles (A-T) was associated with decreased levels of these biomarkers. 

### 3.5. Interactions between Genotypes at the Study Variants and Correlations among the Levels of c-Peptide or Irisin and Other Traits 

Results of examining correlations amongst biomarkers and their interaction with genotypes in the study variants are presented in Table 4. 

The results revealed that (i) measures of c-peptide was mediated by interaction between carrier genotypes (GA + AA) at rs1748197 and measures of IL13; and (ii) measures of irisin were mediated by interaction between carrier genotypes (GA + AA) at rs1748197 and measures of TG. No interaction between the genotypes at the other variant rs12130333 with any of the biomarkers was observed. The model for the interaction of c-peptide with IL13 at the genotypes GG (with intercept coefficient −0.292 and slope 0.26) and (GA + AA) (with mean change 1.52 pg/mL in c-peptide and slope −0.036 (indicated by the sum of 0.261 and −0.2961)) suggested that one unit increase in IL13 would result in 0.036 pg/mL decrease in c-peptide. However, exactly the opposite effect was observed with regard to the reference genotype of GG as evident from the slopes of plot (Figure 2A). The interaction analysis of irisin with TG level at genotypes GG (with intercept 296.58 and slope 66.04) and (GA + AA) (with mean change in irisin −161.75 ng/mL and slope 146.59 (indicated by the sum of 66.04 + 80.55) suggested that one unit increase in TG would result in 146.59 ng/mL increase in irisin (see Table 4). Though a similar effect was observed with genotype GG, steepness of the slope (i.e., effect) at the carrier genotypes (GA + AA) was much higher than that at the reference genotype GG, suggesting that reduction in irisin with increase in TG is more rigorous in individuals carrying the (GA + AA) genotypes (Figure 2B).

### 3.6. Disease Status of the Cohort Participants and the Impact of the Effect Alleles at the Study Variants on the Levels of c-Peptide, Irisin and TG

We performed allele-based logistic regression analysis to evaluate whether the disease status (obesity, diabetes and hypertension) of the study participants is due to the variants (Appendix A). Though the odds-ratio (OR) values were notable, the *p*-values were not significant. This was done to confirm that the impact of the effect alleles in the two variants on the levels of c-peptide, irisin and TG is not due to associated diseases in the study cohort. 

### 3.7. Power Calculation

Power calculation results suggested slight differences between observed (Table 2) and expected (Appendix A) effect sizes for the sample size used for c-peptide (*n* = 161) and irisin (*n* = 217). The observed and expected effect sizes for the variants rs1748197 (MAF = 0.35) and rs12130333 (MAF = 0.15) at closest sample size of c-peptide (*n* = 167) were: observed = −0.69, expected = −0.54 for rs1748197 and observed = −0.90, expected = −0.73 for rs12130333; similarly, the effect sizes considering the closest sample size of irisin (*n* = 214) were observed = −63.1, expected = −54.07 for rs1748197 and observed = −72.61, expected = −72.23 for rs12130333. Similarly, the observed and expected β-values for association of rs1748197 with irisin interacting with TG at sample size of 218 were 80.551 and 91.63, respectively. Furthermore, the observed and expected β-values for association rs1748197 with c-peptide interacting with IL13 at sample size of 160 were −0.296 and −0.113, respectively. At these sample sizes, upon considering type 1 error at >0.05 and population mean ± standard deviation of each of the traits, the observed genetic effect accounts approximately for 3.6% variance in irisin level and 4.6% variance in c-peptide level. 

## 4. Discussion

In our study we investigated two *ANGPTL3* gene variants (rs1748197_A and rs12130333_T) that have been previously associated with metabolite levels and metabolic traits. The rs1748197 variant has been shown to be associated with the metabolites of lipids, particularly those of VLDL and HDL, in a Finnish population [11] while rs12130333 has been associated with TG in cohorts of European and mixed populations [33,34,35] as well as with total cholesterol in a cohort of mixed ancestries [35]. Furthermore, rs1748195, which is in strong LD with rs1748197 (r^2^ = 0.96), has been shown to be associated with TG at genome-wide significance in individuals of European ancestry [36] and had a suggestive *p*-value in a meta-analysis from Kuwait [37]. We report in this study, for the first time, association of these two variants with irisin and c-peptide at significant *p*-values and empirical *p*-values; participants with carrier genotypes at either of the two study variants have significantly lower levels of irisin and c-peptide. Further, haplotype comprising minor alleles of both the SNPs showed significant association with lower levels of irisin. 

Furthermore, we observe suggestive evidence for both the variants to be associated with lower levels of TG. Participants with carrier genotypes at rs1748197 have significantly lower levels of ANGPTL3 protein. It is possible that this reduction in TG level is associated mainly with increased LPL activity since ANGPTL3 levels were also reduced. This could be a main driver in reducing the level of TG seen in people carrying this variant as it has been reported in many other studies that loss of function or oligonucleotide targeting of *ANGPTL3* gene was associated with reduced plasma lipoproteins [38,39]. Oligonucleotide targeting of *ANGPTL3* gene also resulted in reduction of atherogenic lipoprotein levels including LDL-C, VLDL and ApoB, amongst others [38]. Given the fact that ANGPTL3 is a potent inhibitor of LPL [40], it is expected that reducing its level will result in LPL activation. However, interestingly in this paper irisin could also play a role in this pathway. Even though irisin has been initially labelled as a muscle-produced protein, its expression has been also detected in the liver where it played a role in hepatic glucose and lipid metabolism [41]. Additionally, ANGPTL3 is a main regulator of lipid uptake into skeletal muscles through its attenuation of LPL activity, adding another potential mechanism of regulation [42]. As a result, it can be proposed that a crosstalk between the two proteins might mediate their regulation of lipid metabolism potentially through regulating LPL activity. However, further work is needed validate these findings and shed more light on the potential mechanism of action. 

When we tested the effect of type 2 diabetes (T2 D) disease status in the study cohort, we found that the levels of irisin as well as other clinical traits, such as TG, FPG, HbA1c, BMI and waist circumference, were significantly higher in individuals with T2 D when compared to non-diabetic individuals. Levels of irisin are known to be elevated in T2 D and are positively correlated with BMI and negatively correlated with visceral adiposity [43]. Irisin levels are positively associated with insulin resistance as assessed by HOMA- IR [44]. While the higher levels of the above-mentioned biomarkers in the study participants with T2 D are as expected, our results that the effect alleles at the two study variants lead to lower levels of these biomarkers make us inquisitive about the physiological impact of the effect alleles. This is of particular interest especially since our observations indicated that (i) the effect alleles at the two variants do not contribute to the risk of T2 D, obesity or hypertension in our cohort; (ii) we do not find any significant differences in the genotype distributions (carrier versus reference) between diabetic and non-diabetic individuals; (iii) the measurements of anthropometry traits, glycemia traits or lipid traits do not differ significantly between the participants with carrier genotypes and those with reference genotypes; and iv) the effect allele and carrier genotypes at the two study variants are associated with lower levels of irisin and c-peptide in our cohort. 

Our study results suggest that TG mediates the levels of irisin via interaction with the carrier genotypes at rs1748197 and that IL−13 mediates the levels of c-peptide via interaction with the carrier genotypes at rs1748197. In regards to c-peptide, increased levels are observed with the (GA + AA) carrier genotype at rs1748197, while the opposite effect was observed with the reference GG genotype. A reduction in irisin was observed accompanied with an increase in the level of TG. This effect was more evident in individuals carrying the carrier genotypes (GA + AA). Similar results have been previously reported, where serum irisin level was shown to be inversely associated with liver TG contents in obese adults [45]. Irisin is thought to improve hepatic glucose and lipid metabolism by way of positively impacting the liver and pancreatic cells through reducing endoplasmic reticulum stress and thereby acting as an insulin-sensitizing hormone [22]. Studies have demonstrated that high plasma TG is associated with insulin resistance and T2 D [46]. This was also observed among individuals who only show insulin resistance and have not yet developed T2 D [47]. Levels of c-peptide are known to be indicators for MetS and coronary artery disease (CAD) [48,49,50,51,52] and they positively correlate with HOMA-IR [53,54]. Increased level of IL13 has been observed in subjects with insulin resistance [55]. It has been also demonstrated, using mice models, that IL13 plays a key role in the regulation of hepatic glucose production [56] and in controlling blood sugar [57]. Based on the current knowledge as mentioned above on the role of irisin, TG, c-peptide and IL13 and our observation that the effect alleles present at the two study variants lead to lower levels of irisin and c-peptide, we speculate that these two variants provide protection against insulin resistance in people who carry these carrier genotypes. 

Irisin has remained an interesting molecule since its identification as an endurance exercise-induced molecule [25], which was further validated by other studies [58,59]. Nonetheless, it was further reported by Timmons et al. that muscle FNDC5 (the parent protein from which irisin is cleaved) does not change after exercise [27], which was further corroborated in a human clinical trial where no difference was observed in the level of irisin in a group of young healthy males after exercise [59]. These conflicting observations led many people to suggest a problem in the method of quantification [60] while others suggested an alternative mode of regulation for FNDC5/irisin expression in lipid metabolism [41]. Irrespective of the effect of exercise on the irisin level, our data further support a role for irisin in lipid metabolism, potentially through ANGPTL3 and LPL.

We understand that the low sample size of the study cohort as well as the variation in sample size between c-peptide and irisin association studies may slightly inflate the observed effect size of the association analysis. Nevertheless, the strength of these associations was testified by means of empirical *p*-value calculations. Therefore, it is not necessary to test these associations in cohorts of larger sample size.

## 5. Conclusions

The *ANGPTL3* variants are associated with lower levels of irisin and c-peptide and thereby act as protective variants for diabetes and metabolic disorders. The associations between *ANGPTL3* variants and these markers shed light on the possible impact of these variants on metabolism and explain the cardioprotective impact of *ANGPTL3* variants by lowering the level of irisin. Our findings illustrate the key role that the ANGPTL3 plays in connection between lipid metabolism and insulin resistance.

## Figures and Tables

**Figure 1 genes-12-00755-f001:**
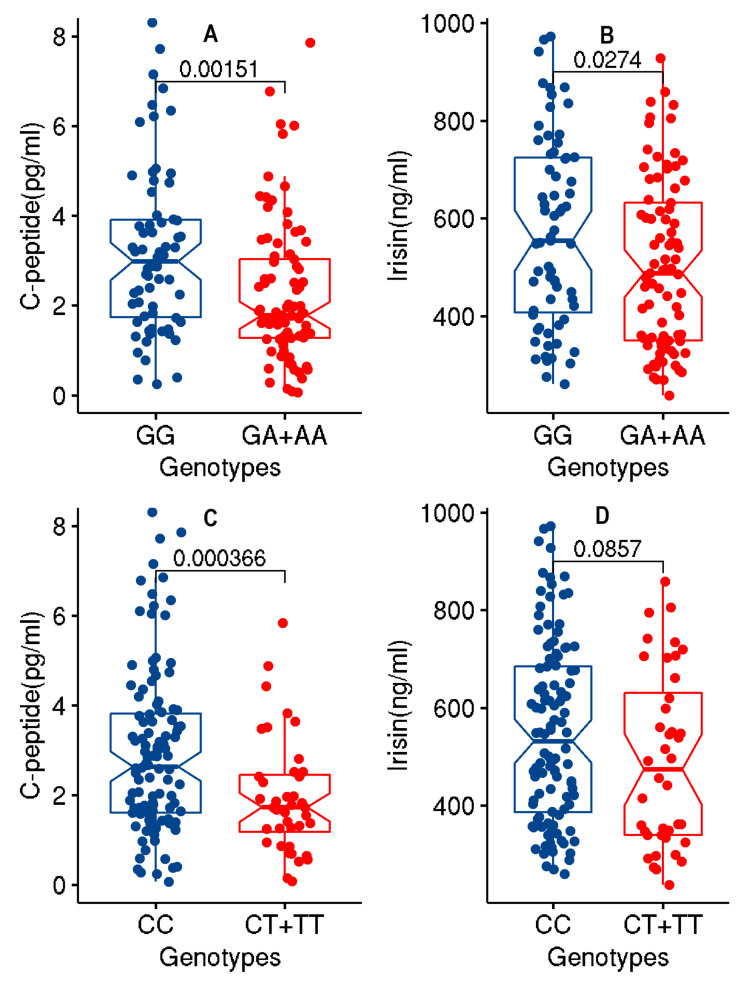
Box plots displaying data distribution for the phenotype traits of c-peptide and irisin in individuals with genotypes (GA + AA) containing the effect allele or genotypes (GG) homozygous for reference allele at variant rs1748197 (Figure 1A,B); and in individuals with genotypes (CT + TT) containing the effect allele or genotypes (CC) homozygous for reference allele at variant rs12130333 (Figure 1C,D).

**Figure 2 genes-12-00755-f002:**
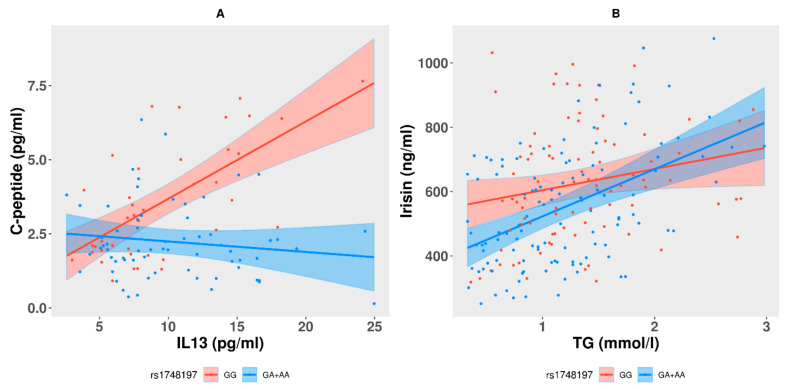
Interactions observed in (**A**) c-peptide with IL13 (negative correlation) and in (**B**) irisin with TG (positive correlation) in interaction with the (GA + AA) carrier genotype at the study variant of rs1748197.

**Table 1 genes-12-00755-t001:** Clinical characteristics of the study cohort. A total of 28 traits and biomarkers were considered in the study.

Traits	Number of Participants Measured	All Participants (Mean ± SD)	Non-Diabetic Participants (Mean ± SD)	Diabetic Participants (Mean ± SD)	*p*-Value ^(a)^ (for Differences in Mean Values between Diabetic and Non-Diabetic Participants)
Male:Female	278	125:153	65:93	60:60	0.1772
Age (in years)	278	46.25 ± 12.38	42.04 ± 12.57	51.8 ± 9.70	3.03 × 10^−12^
Height (in meter)	278	1.64 ± 0.09	1.64 ± 0.09	1.65 ± 0.09	0.516
Weight (in kg)	278	81.40 ± 16.23	77.65 ± 16.49	86.27 ± 14.57	7.07 × 10^−6^
BMI (kg/m^2^)	278	29.93 ± 5.17	28.81 ± 5.46	31.41 ± 4.37	1.36 × 10^−5^
WC (in cm)	177	99.36 ± 13.36	93.95 ± 13.43	105.22 ± 10.57	3.58 × 10^−9^
HDL (in mmol/L)	260	1.20 ± 0.32	1.25 ± 0.31	1.13 ± 0.31	0.0018
TC ^(b)^ (in mmol/L)	272	5.27 ± 1.04	5.56 ± 1.42	5.40 ± 1.22	0.068
LDL ^(b)^ (in mmol/L)	269	3.38 ± 0.94	3.68 ± 1.39	3.51 ± 1.16	0.049
Non-HDL ^(b)^ (in mmol/L)	258	4.018 ± 1.05	4.34 ± 1.31	4.16 ± 1.18	0.031
TG (in mmol/L)	260	1.22 ± 0.59	1.07 ± 0.57	1.43 ± 0.56	7.09 × 10^−7^
FPG (in mmol/L)	241	5.77 ± 1.24	5.21 ± 0.63	6.73 ± 1.42	5.19 × 10^−16^
HbA1 c (%)	254	6.31 ± 1.29	5.61 ± 0.60	7.30 ± 1.36	<2.2 × 10^−16^
Irisin (ng/mL)	219	556.95 ± 192.26	507.96 ± 170.77	620.89 ± 200.67	1.83 × 10^−5^
IL7 (pg/mL) ^(b)^	163	13.23 ± 5.75	12.09 ± 5.44	14.86 ± 5.82	0.0026
IL13 (pg/mL)	158	9.67 ± 4.84	9.21 ± 4.95	10.33 ± 4.63	0.1491
Insulin (pg/mL)	195	14.90 ± 11.97	13.41 ± 11.19	16.83 ± 12.71	0.0511
c-peptide (pg/mL)	161	2.66 ± 1.73	2.76 ± 1.71	2.56 ± 1.75	0.4636
ANGPTL3 (ng/mL)	195	37.42 ± 10.29	36.77 ± 10.33	38.21 ± 10.24	0.3337
TNFa (pg/mL)	167	127.67 ± 32.16	125.31 ± 32.97	131.19 ± 30.81	0.2417
Obese status	278	135:143	62:96	73:47	0.00056
Diabetes status	278	120:158	0:158	120:0	-
Anti-diabetic medication	278	101(med):177 (no med)	158 (no med):0	19 (No med):101 (med)	0.001
Lipid-lowering medication	278	88(med):190	21(med):137	67(med):53	1.43 × 10^−13^

^(a)^ Pearson’s Chi-square test was used for categorical variables and Student’s *t*-test was used for quantitative variables. ^(b)^ Levels of the following 11 interleukins were considered: IL1 b, IL1 ra, IL4, IL5, IL6, IL7, IL8, IL9, IL10, IL13 and IL17. A total of 28 traits and biomarkers were considered in association tests. The values for TC, and LDL were adjusted for lipid-lowering medication by adopting procedures used in [32]: TC_adjusted = TC/0.8; and LDL_adjusted = LDL/0.7. The nonHDL was calculated by subtracting HDL from adjusted TC.

**Table 2 genes-12-00755-t002:** Results of association tests for the two study variants with the levels of c-peptide and irisin, using genetic models based on additive mode of inheritance. Of the 28 traits studied for association with the two variants, only 17 were found independent of one another. Significant *p*-values passing the threshold for multiple testing (*p*-value ≤ 0.003 = 0.05/17) and significant *P**_emp_*-values ≤ 0.05 are highlighted in bold and italic font. Association tests were adjusted for age, sex (regular correction) and further confounders of diabetes medication (DM) and lipid-lowering medication (LLM). Association signals with other traits not meeting the significance dictated by *p*-value threshold adjusted for multiple testing are given in Appendix A.

Traits	SNP with Effect Allele	Correction	Sample Size ^(a)^	β	*p*-Value ^(b)^	Empirical *p*-Value (*P**_emp_*-Value) ^(b)^
c-peptide	rs1748197	R	160	−0.6976	***0.000127***	***0.00679***
		DM	160	−0.6944	***0.000161***	***0.00939***
		LLM	160	−0.6964	***0.000138***	***0.00739***
	rs12130333	R	161	−0.9002	***0.00032***	***0.0154***
		R + DM	161	−0.8991	***0.000335***	***0.0174***
		R + LLM	161	−0.9117	***0.000288***	***0.0167***
Irisin	rs1748197	R	217	−63.1	***0.000299***	***0.0149***
		DM	216	−67.78	***9.58*** × **10^−5^**	***0.0047***
		LLM	216	−63	***0.000357***	***0.0184***
	rs12130333	R	218	−72.61	***0.002135***	0.0979
		R + DM	217	−70.87	***0.002436***	0.1184
		R + LLM	217	−74.3	***0.001806***	0.0898

^(**a**)^ Sample size differs for each of the traits, examination of outliers and missing information for each trait measurement, through quality control steps, led to differing sample sizes for different traits. ^(**b**)^ Significant values are indicated by bold and italics font.

**Table 3 genes-12-00755-t003:** Results of association tests for the haplotype of rs1748197-rs12130333 with the levels of c-peptide and irisin. Effect allele at rs1748197_G_A is A and at rs12130333_C_T is T.

Trait	Haplotype (GC Forms the Reference Haplotype)	Frequency	β	*p*-Value ^(a)^	Empirical *p*-Value (*P**_emp_*-Value) ^(a)^
Irisin	AT	0.134	−74.9	***0.00313***	***0.0122***
	GT	0.013	−110	0.262	0.6224
	AC	0.221	−36.4	0.0951	0.2853
	GC	0.632	68.2	***0.000162***	***0.00059***
c-peptide	AT	0.134	−1.45	***0.0368***	0.1363
	GT	0.013	−2.92	0.304	0.6326
	AC	0.221	−0.49	0.413	0.7832
	GC	0.632	1.17	***0.0182***	0.0891

^(**a**)^ Significant values are indicated in bold and italics font.

**Table 4 genes-12-00755-t004:** Linear regression model illustrating the link between the rs1748197 and interaction of c-peptide with IL13 or of irisin with TG. Results are shown for carrier genotypes (GA and AA) regressed against the reference GG genotype.

Trait (Response variable)	Genotype and Interacting Trait (Predict Variable)	Estimate	Std. Error	*p*-Value	Adj. R-Square	Model *p*-Value
Model:c-peptide~rs1748197 + age + sex + IL13 + rs1748197*IL13
c-peptide	(Intercept)	−0.2927	0.8453	0.7299	0.3416	4.84 × 10^−9^
	rs1748197 (GA + AA)	1.5210	0.6446	0.0202		
	IL13	0.261	0.046	1.96 × 10^−7^		
	rs1748197 (GA + AA):IL13	−0.2961	0.0584	1.88 × 10^−6^		
Model:Irisin~ rs1748197 + age + sex + TG + rs1748197*TG
Irisin	(Intercept)	296.583	74.581	9.78 × 10^−5^	0.2163	2.27 × 10^−10^
	rs1748197 (GA + AA)	−161.75	57.993	0.0058		
	TG	66.04	32.31	0.042		
	rs1748197 (GA + AA):TG	80.551	41.507	0.053

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
