# Peer review of "ANGPTL3 Variants Associate with Lower Levels of Irisin and C-Peptide in a Cohort of Arab Individuals"

_genes, 2021, doi:10.3390/genes12050755_

Round 1

Reviewer 1 Report

This paper by Dr Alanbaei et al. focuses on two ANGPTL3 variants (rs1748197, rs12130333) and their relation to the levels of c-peptide and irisin in a cohort of 278 Arab individuals from Kuwait. ANGPTL3 is at the the time being considered a very relevant therapeutic target in fight against cardiometabolic diseases and several loss-of-function gene variants have been reported to be associated with reduced CVD occurence. The variants in this study have been previously associated with lipid levels especially TGs in very large VLDL particles and TGs in HDL. The authors have specifically investigated the relationships of these ANGPTL3 SNP variants with c-peptide and irisin in order to link possible assocition to a more broad picture, i.e. lipid metabolism, insulin function and possible cardioprotection. However, causality issues are not confirmed in the study, only possible associations. The methods (laboratory, bioinformatics, statistical) are appropriate to test the multitude of aims. The major outcome of the study was that the two ANGPTL3 gene SNP variants associate with irisin and c-peptide significantly and carrier genotypes have significanly lower plasma levels of irisin and c-peptide. In addition, when using haplotypes combining minor alleles of both variants indicated lower levels of Irisin but not c-peptide. Data also suggest that both variants are associated with lower plasma TG concentration. There are several issues that need further discussions.

MAJOR COMMENTS

  1. Since Irisin is highly affected by aerobic physical exercise it is important to have data on the exercise habits of the study participants. Do the authors have this data?
  2. The authors identified only 17 as effective independent traits from the total original of 28 quantitative traits. Why this outcome? Could it be different when using another software than that by Li and Ji 2005?
  3. The authors show their data as gender-wise combined. It would be interesting to see whether there exist any differences between malesd and females. This can be shown in different Table.
  4. The authors do not show what was the method for analyzing plasma ANGPTL3 levels? Show the method and also discuss whether the method measures both full-length and cleaved ANGPTL3 protein forms in plasma since this can cause bias.  Add also information on the c-peptide method.
  5.  The authors suggest that the variants are associated with lower levels of TG. What is the mechanism behind this observation: reduced hepatic production of VLDL, higher LPL activity... Please discuss this issue.
  6. Since low Irisin levels are associated with these ANGPTL3 SNP variants I wonder how the authors interpret their data considering that Irisin is derived from muscles but in muscles mostly ANGPTL4 function is present not ANGPTL3?
  7. The authors claim that IL-13 mediates c-peptide levels via interaction with the carrier genotypes at rs1748197. Where is this interaction demonstrated in this paper? Show the mechanism(s) with all the details included. Same holds with the TG mediation of Irisin levels via interaction with this SNP.

Author Response

We thank the reviewer for the remarks and suggestions. Our answers are detailed bellow:

  1. Since Irisin is highly affected by aerobic physical exercise it is important to have data on the exercise habits of the study participants. Do the authors have this data?

Answer: Unfortunately, this data was not collected from the participants.

  1. The authors identified only 17 as effective independent traits from the total original of 28 quantitative traits. Why this outcome? Could it be different when using another software than that by Li and Ji 2005?

Answer: This outcome helps in deriving P-value threshold (to assess the significance of association signals) corrected for multiple testing and the outcome is not used in any other analysis or data interpretation. The P-value threshold corrected for multiple testing based on all the 28 traits is ≤0.002 (=0.05/28) and that based on effective independent traits is ≤0.002 (=0.05/17). The P-values observed for the reported association signals (Table 2) are ≤0.002.  Thus, there is really no need to delve further into this concern of the reviewer. Further, this particular method is what we have used in our other publications since we have not seen any other software in public domain.

  1. The authors show their data as gender-wise combined. It would be interesting to see whether there exist any differences between malesd and females. This can be shown in different Table.

Answer: We now present a new supplementary table (Table S2 in the line of Table 1) listing the clinical characteristics of the sub-cohorts of only males and of only females. Significant differences were seen in the traits of height, weight, WC, HDL, TG and FPG. We like to point out that the gender-wise differences are taken care in our analysis since the association tests were adjusted for sex (along with age) named as Regular Corrections (R). The following text is now added in the manuscript. (see lines 246-248)

“Examination of the clinical characteristics with participants distributed gender-wise (Supplementary Table S2) showed significant differences in the traits of height, weight, WC, HDL, TG and FPG”.

  1. The authors do not show what was the method for analyzing plasma ANGPTL3 levels? Show the method and also discuss whether the method measures both full-length and cleaved ANGPTL3 protein forms in plasma since this can cause bias.  Add also information on the c-peptide method.

Answer: A detailed ELISA protocol for ANGPTL3 and C-peptide ELISA has been added to the methods section. According to the manufacturing company the ELISA recognizes the recombinant ANGPTL3 protein without specifying which form.

See lines 141-181 for text on ANGPTL3 and c-peptide measurements.

Measurement of plasma levels of ANGPTL3 using ELISA

Plasma level of ANGPTL3 was detected using the quantikine human angiopoietin-like 3 (ANGPTL3) ELISA (R&D systems, Mineapolis, MN, USA. Cat# DANL30). Blood samples from participants were collected using EDTA tubes and plasma was obtained following the centrifugation of the blood tubes at 400xg for 10 min. Plasma was then transferred into new tubes and stored at -80⁰C for later use. For the assay, kit instructions were followed. Briefly; plasma samples were thawed on ice and centrifuged for 5 min at 10,000 ×g at 4°C, to remove any remaining cells or platelets. For the assay samples were diluted 50X with calibrator diluent RD6Q provided in the kit. Standards were prepared in calibrator diluent RD6Q using the recombinant human ANGPLT3 standard provided in the kit (diluted to the following concentrations (ng/mL): 10, 5, 2.5, 1.25, 0.625, 0.313, 0.156). Then 100µl of the assay diluent RD1-76 was transferred into the wells of the plate followed by 50µl of sample and standards into respective wells. The plate was then incubated for 2hrs at room temperature (RT) with gentle shacking. Following incubation, the plate was washed 4X with 1X wash buffer provided in the kit and 200µl of ANGPTL3 conjugated antibody was added into each well. The plate was then incubated for 1hr at RT on shaker, followed by the wash step. The substrate (200µl) was then added into each well and incubated for 30min at RT on benchtop protected from light. Finally, the reaction was stopped using the stop solution provided in the kit and the absorbance of each well was read at 450nm using the Synergy H4 plate reader. Concentration of the unknown samples was determined using the standard curve equation obtained from plotting the standard concentration in relation to the O.D. reading. The intra-assay coefficient for this ELISA assay was 2.0%–5.0%, while the inter-assay coefficient was < 10%.

Measurement of plasma levels of c-peptide using ELISA

C-peptide plasma level was detected using C-peptide ELISA (Mercodia AB, Sylveniusgatan, Sweden, Cat# 10-1136-01). Blood samples from participants were collected, processed, and stored as described in the protocol above. For the assay, kit instructions were followed. Briefly, for the assay samples were centrifuged as described above and used as is. Calibrators and controls used for the assay are provided with the kit in lyophilized form and were reconstituted using distilled water (Calibrator concentrations used were (pmol/L): 95.2, 328, 1270, 2510, 4060). For the assay 25µl of the calibrator, control and sample was transferred into the respective wells of the plate followed by 50µl of assay buffer provided in the kit. The plate was then incubated for 1hr at room temperature (RT) with shacking. Following the incubation, the plate was washed 6X with 1X wash buffer provided in the kit and 100µl of the enzyme conjugate 1X solution, provided in the kit, was added into each well. The plate was then incubated for 1hr at RT on shaker, followed by the wash step. The substrate TMB (200µl) was then added into each well and incubated for 15min at RT on benchtop protected from light. Finally, the reaction was stopped using the stop solution provided in the kit and the absorbance of each well was read at 450nm using the Synergy H4 plate reader. Concentration of the unknown samples was determined using the standard curve equation obtained from plotting the standard concentration in relation to the O.D. reading. The intra-assay coefficient for this ELISA assay was 2.0%–5.0%, while the inter-assay coefficient was < 10%”.

  1.  The authors suggest that the variants are associated with lower levels of TG. What is the mechanism behind this observation: reduced hepatic production of VLDL, higher LPL activity... Please discuss this issue.

Answer: We thank the reviewer for raising this important question. We believe that this reduction in TG level is associated mainly with increased LPL activity since ANGPTL3 levels were also reduced. We have added the following paragraph to further elaborate on this point. The following paragraph is now added to the manuscript text. See lines 370-386.

“It is possible that this reduction in TG level is associated mainly with increased LPL activity since ANGPTL3 levels were also reduced. This could be a main driver in reducing the level of TG seen in people carrying this variant as it has been reported in many other studies that loss of function or oligonucleotide targeting of ANGPTL3 was associated with reduced plasma lipoproteins [PMID: 20942659, PMID: 294543880]. Oligonucleotide targeting of ANGPTL3 also resulted in reduction of atherogenic lipoprotein levels including LDL-C, VLDL and ApoB amongst others [PMID: 32731935]. Given the fact that ANGPTL3 is a potent inhibitor of LPL [PMID: 33011191], it is expected that reducing its level will result in LPL activation. However, interestingly in this paper irisin could also play a role in this pathway. Even though, irisin has been initially labelled as a muscle produced protein, its expression has been also detected in the liver where it played a role in hepatic glucose and lipid metabolism [PMID: 27007446]. Additionally, ANGPTL3 is a main regulator of lipid uptake into skeletal muscles through its attenuation of LPL activity adding another potential mechanism of regulation [PMID: 27053679]. As a result, it can be proposed that a crosstalk between the two proteins might mediate their regulation of lipid metabolism potentially through regulating LPL activity. Nonetheless, further work is needed validate these findings and shed more light on the potential mechanism of action”.

  1. Since low Irisin levels are associated with these ANGPTL3 SNP variants I wonder how the authors interpret their data considering that Irisin is derived from muscles but in muscles mostly ANGPTL4 function is present not ANGPTL3?

Answer: Please refer to our answer to query number 5. In short, irisin is also produced by the liver and ANGPTL3 can function in regulating skeletal muscle lipoprotein uptake through regulating LPL activity.

  1. The authors claim that IL-13 mediates c-peptide levels via interaction with the carrier genotypes at rs1748197. Where is this interaction demonstrated in this paper? Show the mechanism(s) with all the details included. Same holds with the TG mediation of Irisin levels via interaction with this SNP.

Answer: Table 4 and section 3.5 present the data that demonstrate this claim. Please see the column of “genotype and interacting trait” in Table 4 that lists the interacting partners for association signal relating to c-peptide levels and to irisin levels.

Reviewer 2 Report

Dear Editors, 

Thank you for the opportunity to review the manuscript, " ANGPTL3 Variants Associate with Lower Levels of Irisin and C-Peptide in a Cohort of Arab Individuals" by Muath Alanbaei and colleagues. The present study evaluates the impact of two ANGPTL3 common variants (rs1748197 and rs12130333) on the levels of plasma lipids, markers of insulin resistance and irisin in a cohort of 278 Kuwaiti patients. The authors successively found that rs1748197 is associated with lower triglycerides, IL13, c-peptide and irisin plasma levels.

The study is of interest for the cardiometabolic-research community and the manuscript is well written. I have however few major comments and suggestions related to the study.

Major Comment: 

  1. Rs2131925 (in the ANGPTL3 locus) is the top associated SNP with plasma TG levels. In the study the authors genotyped two SNPs (rs1748197 and rs12130333). These two SNPs are in strong LD with the initial one. The use of these two SNPs along the manuscript distracts the readership and, in my view, does not add much to the study (rs1748197 might be enough).
  2. The table 1 presents clinical parameters of the studied population. Moreover, it compares patients with or without T2D. However, the comparison of T2D vs. non-T2D is restricted to clinical parameters. I would be interested in seeing the following genetic associations in these subgroups.
  • For all clinical and biological parameters measured I would like to see the number of patients measured.

minor Comments

  1. Why the authors did not use the rs2131925 (top SNP) as instrument in the cohort ?
  2. It would be interesting to add in the clinical description the LDL-c values corrected for lipid lowering therapies as well as non-HDL cholesterol values.
  • Table 2 and Figure 1 present the same data, I would move the table 2 to supplemental.
  1. I would find interesting to add in the introduction, a brief description of results from clinical trials with Evinacumab and associated clinical challenges.

Author Response

We thank the reviewer for the remarks and suggestions. Our answers are detailed bellow:

  1. Rs2131925 (in the ANGPTL3 locus) is the top associated SNP with plasma TG levels. In the study the authors genotyped two SNPs (rs1748197 and rs12130333). These two SNPs are in strong LD with the initial one. The use of these two SNPs along the manuscript distracts the readership and, in my view, does not add much to the study (rs1748197 might be enough).

Answer: The rs1748197 is in strong LD with the lead SNP at r2=0.995 and the rs12130333 is in moderate LD with the lead SNP at r2=0.5. The reviewer raises a concern that the second study variant rs12130333 does not add much to the study conclusion namely that the ANGPTL3 variant is associated with plasma levels of TG, IL13, c-peptide and irisin. However, it is an interesting variant as it is in the gene region encompassing ANGPTL3, DOCK7 and ATG4C that are associated with TG metabolism. Further, our recent meta-analysis study [PMID: 32902719] found the two ANGPTL3 variants rs1748197 and rs12130333 associated with lipid traits at suggestive P-values – we are taking further these two variants in the current study. Thus, the second variant is still of interest to the cardiometabolic-research community and is probably worth to retain it. Further, the rs1748197 is in perfect LD with the lead SNP at r2=0.995 and the rs12130333 is in moderate LD with the lead SNP at r2=0.5. Thus, it is interesting to study how these two variants (one almost as good as the lead SNP and the other with moderate LD) behave in terms of association with metabolic traits and markers.

  1. The table 1 presents clinical parameters of the studied population. Moreover, it compares patients with or without T2D. However, the comparison of T2D vs. non-T2D is restricted to clinical parameters. I would be interested in seeing the following genetic associations in these subgroups.

Answer: We appreciate this suggestion. However, we like to point out that the sub-cohorts derived by the diabetes status can become small as irisin was measured only in 216 patients and c-peptide in 160 patients in the study cohort. Considering the small sizes of the sub-cohorts, results from association tests with the two sub-cohorts may be considered with caution. The summary statistics for association tests with c-peptide and irisin with the two sub-cohorts are given in Table S5. The P-values for associations of the two variants with levels of irisin and c-peptide are consistently significant in the sub-cohort of diabetes individuals. We now add the following text in the manuscript. See lines 267-290.

“Evaluation of the identified association signals from these two variants with the levels of irisin and c-peptide revealed that these two association signals were consistently significant (P-value≤0.05) in the sub-cohorts of individuals afflicted with diabetes (Supplementary Table S5)”.

  • For all clinical and biological parameters measured I would like to see the number of patients measured.

Answer: We have now revised Table 1 by way of giving the number of patients measured for each of the listed traits.

minor Comments

  1. Why the authors did not use the rs2131925 (top SNP) as instrument in the cohort ?

Answer: Our recent meta-analysis study [[PMID: 32902719] found the two ANGPTL3 variants rs1748197 and rs12130333 associated with lipid traits at suggestive P-values. In the current study, we take further the above observation.

The rs1748197 is in strong LD with the lead SNP at r2=0.995 and the rs12130333 is in moderate LD with the lead SNP at r2=0.5. While 20 studies are listed in GWAS Catalog reporting association signals with the lead SNP, only 2 studies for rs1748197 and 4 studies for rs12130333 are listed. Thus, it is interesting to study how these two variants (one almost as good as the lead SNP and the other with moderate LD) behave in terms of association with metabolic traits and markers.   

  1. It would be interesting to add in the clinical description the LDL-c values corrected for lipid lowering therapies as well as non-HDL cholesterol values.

Answer: We now adjusted the LDL and TC values for lipid lowering medication by adopting procedures used in [PMID: 29083408]: TC_adjusted=TC/0.8; and LDL_adjusted=LDL/0.7. The nonHDL was calculated by subtracting HDL from adjusted TC.  Such adjusted values are now presented in Table 1 and Table S2. Further, association tests were performed with such adjusted values (see Supplementary Table S4).

  1. Table 2 and Figure 1 present the same data, I would move the table 2 to supplemental.

Answer: Agreed. Moved Table 2 to supplement information as Supplementary Table S3.

  1. I would find interesting to add in the introduction, a brief description of results from clinical trials with Evinacumab and associated clinical challenges.

Answer: Yes, it is interesting to mention Evinacumab. We have now added the following paragraph in the manuscript text. See lines 83-93.

“Based on data emerging from various GWA studies on ANGPTL3 and its role in regulating plasma lipid levels, ANGPTL3 has been targeted as a treatment for people with elevated plasma lipids, especially for people diagnosed with homozygous familial hypercholesterolemia. Evinacumab, the ANGPTL3 inhibitor, is a human monoclonal antibody that has been recently approved by United States Food and Drug Administration (FDA).  In the Phase 3 clinical trial, patients receiving Evinacumab showed a 49% reduction in LDL-C levels from baseline relative to those receiving placebo at week 24 after treatment [PMID: 32813947]. Indeed, this result is very promising since LDL reduction in patients with homozygous familial hypercholesterolemia is very difficult [PMID: 30586774]. However, the trial assessed only a small number of patients for a limited period where long term safety and cardiovascular outcomes risk have not been fully examined [PMID: 32813947]”.

Reviewer 3 Report

Teh manuscript of Alanbaei et al. reports the association of two SNPs, rs1748197 and rs121303333, with various plasma cocnentrations of metabolism-associated proteins. I have several concerns with respect to this manuscript.

  1. First of all, the authors report that both SNPs are ANGPTL3 variants. But do they also affect plasma ANGPTL3 and/or ANGPTL3 activity? This is not clear at all. It is mentioned that the rs1748197 variant affects ANGPTL3, but there is no effect of teh other SNP? Only the rs1748197 is relatively close to the ANGPTL3 gene. Might it be that the rs12130333 is not a 'ANGPTL3 SNP'?
  2. How do the authors think that the liver-specific protein ANGPTL3 affects the muscle-specific irisin protein? This is still not clear to me, despite the rather long discussion.
  3.  There is much debate about the presence of an (active) irisin protein in the human circulation. See for instance Albrecht et al (PMID 32180552). Please comment on this; is there indeed an effect on irisin? 
  4. Missing form the Materials and Methods section: How did the researcher determien the various plasma parameters? Only detail on irisin measurement are provided. 
  5. Please use similar Y-axis in theFigure 1A and 1C enabling readers to compare the Figures.

Author Response

We thank the reviewer for the remarks and suggestions. A detailed answer is given bellow.

  1. “First of all, the authors report that both SNPs are ANGPTL3 variants. But do they also affect plasma ANGPTL3 and/or ANGPTL3 activity? This is not clear at all. It is mentioned that the rs1748197 variant affects ANGPTL3, but there is no effect of teh other SNP? Only the rs1748197 is relatively close to the ANGPTL3 gene. Might it be that the rs12130333 is not a 'ANGPTL3 SNP'?

Answer: Since both the variants are non-coding, they are not expected to affect the activity of ANGPTL3.  As regards the impact on the levels of ANGPTL3, we found that (i) there is significant difference in the plasma level of ANGPTL3 between individuals carrying reference genotype as opposed to individuals carrying carrier genotypes at rs1748197; and (ii) the rs1748197 variant is significantly associated with ANGPTL3 levels. Such observations are not seen with rs12130333 in the current cohort. These lead the reviewer to ask the question of whether it is an ANGPTL3 variant. However, a number of reports in literature presents rs12130333 as located within the ANGPTL3 locus (i.e. ANGPTL3 SNP) or as near ANGPTL3 gene [e.g. PMID: 22896670; PMID: 18596051; PMID: 21691831; PMID: 29334984; PMID: 22659251]. Thus, we feel that we can retain this SNP in the manuscript.  

The following text is now provided in the manuscript. See lines 63-66.

The rs1748197 is an intronic SNP from ANGPTL3 gene and the rs12130333 is a regulatory region variant downstream of ANGPTL3 gene. A number of reports in literature presents rs12130333 as located within the ANGPTL3 locus (i.e. ANGPTL3 SNP) or as near ANGPTL3 gene [e.g. PMID: 22896670; PMID: 18596051; PMID: 21691831; PMID: 29334984; PMID: 22659251]”.

  1. How do the authors think that the liver-specific protein ANGPTL3 affects the muscle-specific irisin protein? This is still not clear to me, despite the rather long discussion.

Answer: We thank the reviewer for this critical question. The following paragraph has now been added to the discussion (see lines 379-386). In short, irisin is also produced by the liver and ANGPTL3 can function in regulating skeletal muscle lipoprotein uptake through regulating LPL activity.

“Even though, irisin has been initially labelled as a muscle produced protein, its expression has been also detected in the liver where it played a role in hepatic glucose and lipid metabolism [PMID: 27007446]. Additionally, ANGPTL3 is a main regulator of lipid uptake into skeletal muscles through its attenuation of LPL activity adding another potential mechanism of regulation [PMID: 27053679]. As a result, it can be proposed that a crosstalk between the two proteins might mediate their regulation of lipid metabolism potentially through regulating LPL activity. However, further work is needed validate these findings and shed more light on the potential mechanism of action”.

”.

  1.  There is much debate about the presence of an (active) irisin protein in the human circulation. See for instance Albrecht et al (PMID 32180552). Please comment on this; is there indeed an effect on irisin? 

Answer: This is a good question and for sure further experimentation will be needed to fully answer this question. At this stage we strongly believe that this is a true observation based on the observed role of irisin in lipid metabolism as we suggest below. The following paragraph has now been added (see lines 436-444.

“Irisin has remained an interesting molecule since its identification as an endurance exercise induced molecule [PMID: 22237023], which was further validated by other studies [PMID: 28944087, PMID: 23018146]. Nonetheless, it was further reported by Timmons et al that muscle FNDC5 (the parent protein from which irisin is cleaved) was not changing after exercise [PMID: 22932392] which was further corroborated in a human clinical trial where  no difference was observed in the level of irisin in a group of young healthy males after exercise [PMID: 23018146]. These conflicting observations led many people to suggest a problem in the method of quantification [PMID: 32180552] while others suggested an alternative mode of regulation for FNDC5/irisin expression in lipid metabolism [PMID: 27007446]. Irrespective of the effect of exercise on the irisin level, our data further supports a role for Irisin in lipid metabolism potentially through ANGPTL3 and LPL”.

  1. Missing form the Materials and Methods section: How did the researcher determine the various plasma parameters? Only detail on irisin measurement are provided. 

Answer: A detailed ELISA protocol for ANGPTL3 and C-peptide ELISA has now been added to the methods section. See lines 141-181.

Measurement of plasma levels of ANGPTL3 using ELISA Plasma level of ANGPTL3 was detected using the quantikine human angiopoietin-like 3 (ANGPTL3) ELISA (R&D systems, Mineapolis, MN, USA. Cat# DANL30). Blood samples from participants were collected using EDTA tubes and plasma was obtained following the centrifugation of the blood tubes at 400xg for 10 min. Plasma was then transferred into new tubes and stored at -80⁰C for later use. For the assay, kit instructions were followed. In brief, plasma samples were thawed on ice and centrifuged for 5 min at 10,000 ×g at 4°C, to remove any remaining cells or platelets. For the assay samples were diluted 50X with calibrator diluent RD6Q provided in the kit. Standards were prepared in calibrator diluent RD6Q using the recombinant human ANGPLT3 standard provided in the kit (diluted to the following concentrations (ng/mL): 10, 5, 2.5, 1.25, 0.625, 0.313, 0.156). Then 100µl of the assay diluent RD1-76 was transferred into the wells of the plate followed by 50µl of sample and standards into respective wells. The plate was then incubated for 2hrs at room temperature (RT) with gentle shacking. Following incubation, the plate was washed 4X with 1X wash buffer provided in the kit and 200µl of ANGPTL3 conjugated antibody was added into each well. The plate was then incubated for 1hr at RT on shaker, followed by the wash step. The substrate (200µl) was then added into each well and incubated for 30min at RT on benchtop protected from light. Finally, the reaction was stopped using the stop solution provided in the kit and the absorbance of each well was read at 450nm using the Synergy H4 plate reader. Concentration of the unknown samples was determined using the standard curve equation obtained from plotting the standard concentration in relation to the O.D. reading. The intra-assay coefficient for this ELISA assay was 2.0%–5.0%, while the inter-assay coefficient was < 10%.

Measurement of plasma levels of c-peptide using ELISA

C-peptide plasma level was detected using C-peptide ELISA (Mercodia AB, Sylveniusgatan, Sweden, Cat# 10-1136-01). Blood samples from participants were collected, processed, and stored as described in the protocol above. For the assay, kit instructions were followed. Briefly, for the assay samples were centrifuged as described above and used as is. Calibrators and controls used for the assay are provided with the kit in lyophilized form and were reconstituted using distilled water (Calibrator concentrations used were (pmol/L): 95.2, 328, 1270, 2510, 4060). For the assay 25µl of the calibrator, control and sample was transferred into the respective wells of the plate followed by 50µl of assay buffer provided in the kit. The plate was then incubated for 1hr at room temperature (RT) with shacking. Following the incubation, the plate was washed 6X with 1X wash buffer provided in the kit and 100µl of the enzyme conjugate 1X solution, provided in the kit, was added into each well. The plate was then incubated for 1hr at RT on shaker, followed by the wash step. The substrate TMB (200µl) was then added into each well and incubated for 15min at RT on benchtop protected from light. Finally, the reaction was stopped using the stop solution provided in the kit and the absorbance of each well was read at 450nm using the Synergy H4 plate reader. Concentration of the unknown samples was determined using the standard curve equation obtained from plotting the standard concentration in relation to the O.D. reading. The intra-assay coefficient for this ELISA assay was 2.0%–5.0%, while the inter-assay coefficient was < 10%”.

  1. Please use similar Y-axis in the Figure 1A and 1C enabling readers to compare the Figures.

Answer: Many thanks for pointing out this discrepancy, we have now redrawn the Figure 1A and 1C with similar Y-axis.

Round 2

Reviewer 1 Report

The authors have responded in a satisfactory manner to my queries. There still remains on detail that should be corrected as follows:

In Discussion the authors have two times the same text (i.e. lines 379-386 and 427-434). Please correct this duplicate information to one. 

Author Response

We thank the reviewer for finding the revision satisfactory.

We have now removed, in track mode, the second occurrence of the duplicated text.

We also carried out spell-checks.

Reviewer 2 Report

I thank the authors answering all my previous concerns. 

Author Response

We thank the reviewer for finding our revision satisfactory.

We carried out spell-check.

Reviewer 3 Report

The authors have addressed my comments satisfactorily. I would however suggest to shorten the description of the ANGPTL3 and c-Peptide ELISAs. It will be enough to just mention the kit (name/manufactor) and that it was used according to the manufactory protocol.